# Probing the Dynamics of Li^+^ Ions on the Crystal Surface: A Solid-State NMR Study

**DOI:** 10.3390/polym12020391

**Published:** 2020-02-09

**Authors:** Bi-Heng Wang, Tian Xia, Qun Chen, Ye-Feng Yao

**Affiliations:** Material Science Department & Physics Department & Shanghai Key Laboratory of Magnetic Resonance, School of Physics and Electronic Science, East China Normal University, North Zhongshan Road 3663, Shanghai 200062, China; wangbh0912@foxmail.com (B.-H.W.); txia@phy.ecnu.edu.cn (T.X.); qchen@ecnu.edu.cn (Q.C.)

**Keywords:** solid-state NMR, polymer electrolytes, Li^+^ motion

## Abstract

Polyethylene oxide-based solid polymer electrolytes (SPEs) are of research interest because of their potential applications in all-solid-state Li^+^ batteries. However, despite their advantages in terms of compatibility with the electrodes and easy processing, polyethylene oxide (PEO)/Li^+^ complexes often suffer from low conductivity at room temperature. Understanding the conduction mechanism and, in turn, developing strategies to improve the conductivity have long been the main objectives underlying research into PEO/Li^+^ complex electrolytes. Here, we prepared several special PEO/Li^+^ complex samples where the PEO/Li^+^ complex structures were located on the surfaces of PEO crystals and consisted of high content chain ends. We found two different Li^+^ species in the PEO/Li^+^ complex structures via solid-state nuclear magnetic resonance (NMR). The 2D ^7^Li exchange NMR showed the exchange process between the different Li^+^ species. The exchange dynamics of the Li^+^ ions provide a molecular mechanism of the Li^+^ transportation in the surface of PEO crystal lamella, which is further correlated with the ionic conduction mechanism of the PEO/Li^+^ complex structure.

## 1. Introduction

The polyethylene oxide/Li^+^ (PEO/Li^+^) complex is a solid polymer electrolyte (SPE) with many potential applications in all-solid-state Li^+^ batteries [1,2,3,4]. However, the relatively low conductivity of PEO/Li^+^ complexes at room temperature is the Achilles’ heel of these materials. Understanding the conduction mechanism and, in turn, developing the strategies to improve the material’s conductivity have long been the main objectives of research into PEO/Li^+^ complex SPE [5,6].

On the molecular level, the conduction of PEO/Li^+^ complexes has been correlated with Li^+^ motions [7,8], which are coupled with segmental motions of the PEO chain due to the coordination between Li^+^ ions and PEO chain segments [1,9,10,11]. The chain ends can initiate and facilitate the segmental motions of polymer chain [12,13,14,15] and are thus expected to have a strong influence on the motions of the Li^+^ ions coordinated to the polymer chains [16,17,18]. PEO/Li^+^ complex materials have a high mobility of the chain ends, which strongly influences the Li^+^ transport and thus the material conductivity [19,20]. However, the Li^+^ motion is only elemental. The local processes of the Li^+^ ions in the PEO/Li^+^ complexes, as well as their conduction require long-range transport of Li^+^ ions [21,22]. Considering the sample morphology, the ionic conduction of the PEO/Li^+^ complex is more complicated than what was described above.

For example, most PEO/Li^+^ complexes are semi-crystalline [23]. In semi-crystalline polymers, the crystallites are always surrounded by non-crystalline structures [24,25,26]. Such morphologies are also present in most PEO/Li^+^ complexes. Moreover, it is known that for most semi-crystalline SPEs, the conductivity of non-crystalline structures is often higher than that of crystalline ones [27,28,29,30]. Li et al. demonstrated that the conductivity of the non-crystalline phase in the surface of the PEO/Li^+^ complex single crystal was 2–3 orders of magnitude greater than that of the through-plane phase [31]. In semi-crystalline PEO/Li^+^ complexes, it is possible that the conduction process may proceed in the highly conductive non-crystalline components on the crystal surface rather than passing through the low conductive crystals. In this context, conduction on the crystalline surface seems to be critical to our understanding of the conduction mechanism of semi-crystalline PEO/Li^+^ complexes.

Here, we report a detailed solid-state NMR study on the dynamics of the Li^+^ ions on the crystalline surface. The PEO/Li^+^ complex samples were prepared by immersing PEO flakes in the Li salt solvents [31]. By controlling the immersion time, the Li^+^ ions in the solvents can gradually permeate the surface portions of the PEO crystal lamellae while the PEO crystal cores are maintained. This preparation procedure can therefore ensure that the Li^+^ ions are only located on the surfaces of the crystals. Meanwhile, the surface areas of PEO lamella contain a high content of chain ends, and thus, these samples offer a good opportunity to probe the structure and dynamics of the Li^+^ ions around the chain ends; this is usually quite difficult in routinely prepared PEO/Li^+^ complex samples. The use of solid-state NMR techniques revealed two different Li^+^ species in the PEO/Li^+^ complex structures. The 2D ^7^Li exchange NMR demonstrated that the Li^+^ ions around the chain ends could exchange with those coordinated with the amorphous chain segments. The exchange dynamics identified here offer a molecular mechanism for Li^+^ transportation on the surface of a PEO crystal lamella, which is further correlated to the ionic conduction mechanism of the PEO/Li^+^ complex material.

## 2. Materials and Methods

### 2.1. Sample Preparation

All the chemical reagents were purchased from Sigma-Aldrich (Sigma-Aldrich Inc., St. Louis, MO, USA). The polyethylene oxide (PEO) had a molecular weight (*M*_w_) of 1500. Previous studies showed that such a low molecular weight PEO generally forms crystals with a fully-extended chain, i.e., the fully extended-chain crystals [32,33,34,35]. To prepare these samples, PEO powders were first pressed to form flakes with a diameter of 0.75 cm and a thickness of 200 μm. The flakes were then immersed in the LiCF_3_SO_3_ pentyl acetate solution. PEO has a very low solubility in LiCF_3_SO_3_ pentyl acetate solution, and the flakes do not dissolve. However, during the immersion process, Li^+^ ions may interact with the polymer chains on the PEO crystal surface. The Li^+^ ions can gradually diffuse into the crystal surface and gradually penetrate the crystal because of the coordination interactions between Li^+^ ions and oxygen atoms of PEO chain. This leads to a decrease in the lamellar thickness (Figure 1). We prepared three samples with different immersion times (105, 240, 390 min); these samples were named IM105-PEO/Li^+^, IM240-PEO/Li^+^, and IM390-PEO/Li^+^, respectively. After immersion, the samples were dried under vacuum at 313 K for 5 days to remove the solvent. We then used ^7^Li solid-state NMR to measure the Li^+^ contents in the samples (see Appendix A). We calculated the ratios of Li^+^/EO in these samples based on the Li^+^ contents (Table 1).

For comparison, a (PEO)_3_LiCF_3_SO_3_ complex crystal sample was prepared. The protocol used PEO (average *M*_w_ = 1500) and LiCF_3_SO_3_ dried in a vacuum for 24 h. The PEO and LiCF_3_SO_3_ were mixed with a molar ratio EO_PEO_:Li^+^ of 3:1. These were dissolved in acetonitrile and continuously stirred for 24 h at 293 K. The solution was then cast on a Teflon plate. After complete evaporation of the acetonitrile, the sample was further dried in a vacuum for a week at 313 K before the measurements.

### 2.2. Solid-state NMR Experiments

All the ^7^Li and ^13^C MAS NMR experiments were performed on a Bruker AVANCE III 600 WB spectrometer (Bruker Inc., Karlsruhe, Germany) operating at 233.23 and 150.11 MHz for ^7^Li and ^13^C, respectively. In the ^7^Li experiments, the recycle delay was set to 150 s to ensure complete relaxation of the ^7^Li spins. The exchange time was set to 1–200 ms in the 2D ^7^Li–^7^Li exchange experiments; the MAS speed was 10 kHz. The static ^7^Li quantitative experiments were performed on a Bruker AVANCE III 400WB spectrometer operating at 155.52 MHz for ^7^Li. The ^7^Li and ^13^C chemical shifts were calibrated using LiCl aqueous solution (1 mol/L, 0 ppm) and adamantine (38.56 ppm), respectively.

We designed a unique pulse sequence for ^13^C single pulse excitation NMR to monitor the ^13^C amorphous signals. This sequence used a very short relaxation delay to recover the desired amorphous signals selectively (see Figure 2). The train of the π/2 pulses saturated the signals recovered during the recycling delay. The following relaxation delay was then used to recover the desired signals selectively with a very fast T_1_ relaxation (spin lattice relaxation). The relaxation delay was set to 0.05 s to select the mobile amorphous signals.

### 2.3. X-ray Diffraction Measurements

The X-ray diffraction measurements were performed on a Rigaku ULTIMA IV (Rigaku Inc., Tokyo, Japan) using Cu-Ka (1.5406 Å) radiation (35 kV, 25 mA). All samples were mounted on a same sample holder and scanned from 2θ = 5°–50° at a speed of 10°/min. These experiments were then performed at room temperature.

### 2.4. Differential Scanning Calorimetry

The DSC measurements were from a DSC Q2000 series instrument (TA Instruments, New Castle, Germany). Approximately 1 to 3 mg of the SPE samples were tested over a temperature range of 10 to 180 °C at a scanning rate of 10 °C/min.

### 2.5. Electrochemical Impedance Spectroscopy

The EIS measurements were performed using an electrochemical workstation (PARSTAT 4000A, Ametek Scientific Instruments, Berwyn, PA, USA). The SPE flakes were sandwiched between two stainless steel plates in a two electrode cell placed in an argon-filled stainless steel chamber.

## 3. Results and Discussion

### 3.1. The Structure of IM-PEO/Li^+^

We prepared three samples by controlling the immersion time. The immersion time of these samples varied from 105 to 390 min. The sample with an immersion time of 240 min (IM240-PEO/Li^+^) was then used for a detailed investigation. Detailed characterization of the other samples is in the Appendix A (see Appendix A). For a comparison, an untreated PEO flake sample (neat PEO) was chosen in the study. By combining the static quantitative ^7^Li NMR experiment and DSC experiments, we determined the ratio of EO:Li^+^ in the amorphous phase of IM240-PEO/Li^+^. The ratio of EO:Li^+^ in the amorphous phase of this sample was 2.4:1. A detailed determination procedure is given in the experimental section and Appendix A.

Figure 3b shows the WAXD patterns of the neat PEO, IM240-PEO/Li^+^, and (PEO)_3_LiCF_3_SO_3_. In the WAXD pattern of PEO, the peaks at 2θ = 19.16° and 23.33° were the characteristic reflection peaks of the PEO crystal; they could be assigned to the crystallographic planes of (120) and (032), respectively. The WAXD patterns of IM240-PEO/Li^+^ showed very similar diffraction peaks, indicating that the dominant crystal structure in IM240-PEO/Li^+^ was the PEO crystals. The diffraction peaks of the (PEO)_3_LiCF_3_SO_3_ complex crystal were missing in the pattern, indicating that the PEO/Li^+^ complexes were completely amorphous. The presence of the PEO crystal and the missing (PEO)_3_LiCF_3_SO_3_ complex crystals in IM240-PEO/Li^+^ were further confirmed by the DSC measurements of the sample (see Figure 3c) [32]. Low molecular weight PEOs, like the PEO sample (*M*_w_ = 1500) used here, are prone to forming fully extended chain crystals [33,34,35]. In this sample morphology, Li^+^ ions will first interact with the non-crystalline structures on the crystal surface when the crystals are immersed into a solvent. With increasing immersion time, the Li^+^ ions can be enriched on the surface and, in turn, even diffuse into the PEO crystalline, resulting in reduced crystallinity (Figure 3a).

Figure 4a shows the ^13^C single pulse excitation spectra of neat PEO and IM240-PEO/Li^+^. The signals in the spectra were assigned according to the literature [36,37]. In the spectrum of neat PEO, the broad peak centered at 72.5 ppm was assigned to the crystalline signal, and the narrow peak centered at 71.2 ppm was the amorphous signal. The small and narrow signal centered at 61.8 ppm was assigned to the signal from the chain ends [38]. The signals in the spectrum of IM240-PEO/Li^+^ were quite similar to those of neat PEO. The signal at 72.5 ppm in the spectrum of IM240-PEO/Li^+^ could also be assigned to the PEO crystals. Compared to the neat PEO, this signal showed a clear decrease in the intensity indicating a decrease in the sample crystallinity. The narrow signal centered at 70.7 ppm was assigned to the amorphous PEO/Li^+^ complex structures. Compared to the amorphous signal of neat PEO, this signal showed a slight up-field shift that was attributed to the formation of the coordination structures between the Li^+^ ions and PEO segments. The signal at 61.8 ppm was the signal of the chain ends [38]. This signal showed a clear increase in the linewidth, indicating a reduction in the chain mobility. This was likely due to the formation of complex structures between the chain ends and the Li^+^ ions, which will be discussed in detail below.

### 3.2. The Li^+^ Ions Around the Amorphous Chain Segments

Figure 4b shows the ^7^Li single pulse excitation spectrum of IM240-PEO/Li^+^. For comparison, the spectrum of (PEO)_3_LiCF_3_SO_3_ was also acquired using the same experimental conditions (Figure 4b). The ^7^Li spectrum of (PEO)_3_LiCF_3_SO_3_ had small signals centered at −0.91 ppm; these were assigned to the Li^+^ ions in the amorphous phase of (PEO)_3_LiCF_3_SO_3_. The signal centered at −1.20 ppm could be assigned to the Li^+^ ions in the crystal phase (see Appendix A and the related discussion). The ^7^Li NMR spectrum of IM240-PEO/Li^+^ had two narrow signals centered at −0.78 and −0.91 ppm, indicating the presence of two different Li^+^ species in the sample. By comparing the ^7^Li signals of (PEO)_3_LiCF_3_SO_3_ with those of IM240-PEO/Li^+^, we assigned the signal at −0.91 ppm to the Li^+^ ions in the amorphous PEO/Li^+^ complex structures of IM240-PEO/Li^+^ [39,40,41]. The signal at −0.78 ppm had not yet been reported to the best of our knowledge. Considering the structural features of IM240-PEO/Li^+^ and the spatial locations of the Li^+^ ions, we tentatively assigned this signal to the Li^+^ ions around the chain ends (most probably coordinated with the chain ends). More detailed discussion about this signal assignment is presented below.

### 3.3. The Li^+^ Ions Around the Chain Ends

Figure 5a shows the temperature-dependent ^7^Li NMR spectra of IM240-PEO/Li^+^. The intensity of the signal at −0.78 ppm decreased with increasing temperature while the signal at −0.91 ppm showed a clear increasing intensity. These changes in the spectra strongly indicated that the Li^+^ ions corresponded to the two signals and may transform each other. Increasing temperature could facilitate the formation of the Li^+^ species corresponding to the signal at −0.91 ppm, whereas the Li^+^ species corresponding to the signal at −0.78 ppm were more stable at low temperatures.

Figure 5b shows the temperature-dependent ^13^C single pulse excitation spectra of IM240-PEO/Li^+^. To selectively probe the amorphous signals, the spectra in Figure 5b were acquired using the modified single excitation pulse sequence with a short relaxation delay of 0.05 s (see the pulse sequence scheme in Figure 2). At 285 K, the amorphous signal had a wide peak centered at 70.7 ppm. With increasing temperature, this signal clearly decreased in terms of signal width. This signal narrowing could be attributed to the increased segmental mobility at elevated temperatures. However, the chemical shift of the amorphous signal remained nearly unchanged within the temperature range, indicating that the conformation statistics of the chain segments remained nearly constant in the amorphous phase. Figure 5b also presents the temperature-dependent ^13^C signal of the chain ends. At 285 K, the ^13^C signal of the chain ends appeared as a wide hump centered at 61.4 ppm. With increasing temperature, this signal showed a clear down-field shift accompanied by a gradual decrease in line width. The chemical shift change indicated that the conformational statistics of the chain ends changed with increasing temperature. The decrease in line width indicated the increase in mobility of the chain ends.

These observations helped explain the origin of the ^7^Li signals at −0.78 ppm in Figure 5a. The signals at −0.78 ppm in the ^7^Li spectra showed a clear decrease with increasing temperature, thus indicating the decreasing amount of Li^+^ species. Meanwhile, the ^13^C NMR in Figure 5b showed that the chain ends had a clear downfield shift towards the ^13^C chemical shift value of neat amorphous PEO with increasing temperature. This suggested that the PEO chain ends in IM240-PEO/Li^+^ formed conformational structures close to those in neat amorphous PEO with increasing temperature. The coordination structures between Li^+^ ions and the chain ends in IM240-PEO/Li^+^ became loose or even dissociated with increasing temperature. This change agreed with the Li^+^ (−0.78 ppm) data in ^7^Li NMR: The Li^+^ species (−0.78 ppm) were strongly associated with the chain ends.

The states of the anions, CF_3_SO_3_^−^, in the sample were an interesting question. To have an electrostatic equilibrium, the anions must also get into the crystal surface regions together with the Li^+^ ions. To probe the states of the anions, we collected the ^19^F NMR spectrum of IM240-PEO/Li^+^ (see Appendix A). It was found that only one signal appeared in the spectrum, indicating that the anions only had one state in the sample. However, the detailed study of the anions was beyond the scope of this work. Further experimental study of the states of the anions as well as the interaction between the anions and the Li^+^ ions was the subject of ongoing studies in our laboratory.

### 3.4. The Local Dynamics of Li^+^ Ions in IM240-PEO/Li^+^

Next, 2D ^7^Li–^7^Li exchange NMR was used to probe the local exchange dynamics between the Li^+^ ions coordinated with PEO segments (−0.91 ppm) and the Li^+^ ions coordinated with the PEO chain ends (−0.78 ppm). Figure 6a shows the 2D ^7^Li–^7^Li exchange spectrum acquired using the exchange time varying from 1 to 200 ms. The experimental temperature was 305 K. Clear cross-peaks were observed in the spectrum, indicating the presence of the exchange dynamics between the two Li^+^ species. To quantify the exchange rate, a two site exchange (e.g., Li-1 ↔ Li-2) was introduced to model the process [42]. Based on this model, the exchange rate, *k*, could be calculated from the ratio between the cross-peak intensity and the diagonal peak intensity:(1)$$\frac{a_{cross}^{1\to 2}}{a_{diag}^{1}}=\frac{e^{2k\tau }-1}{e^{2k\tau }+1}\times \exp\left(-\frac{\tau }{T^{*}}\right)$$

Here, *a* refers the intensity of the cross-/diagonal peak and *τ* is the exchange time. The term *T** is the apparent relaxation time [42]. 

In the 2D exchange of the ^7^Li-^7^Li spectra of IM240-PEO/Li^+^, the cross-section was extracted at −0.78 ppm with an exchange time of 10 ms. This ensured that only a single step jump occurred. The integral area ratios of the cross-peak and the diagonal peak are listed in Figure 6b. Figure 6c is the fitting curve between the ratio of the cross-/diagonal peak and the exchange time. The curve was fitted according to Equation (1), and the resulting exchange rate was 38 s^−1^ at 305 K. We then used this approach to measure the exchange rates of the Li^+^ ions in IM240-PEO/Li^+^ from 280 to 305 K (see Figure 7a).

Figure 7a gives the Arrhenius plots of the exchange rates. The fitting yielded an activation energy of 28.7 ± 2.3 kJ/mol. This activation energy value was much lower than the PEO/Li^+^ complex crystals [43] and even the amorphous PEO/Li^+^ complexes [44], thus indicating very weak coordination between the Li^+^ ions and the ligands (i.e., the PEO chain segments and the chain ends). This weak coordination could be associated with the high mobility of the amorphous chain segments in the sample, especially the high mobility of the chain ends. We tried to determine the connection between this exchange process and ionic conductivity in this study. 

Li et al. previously studied the conductivity of the crystal surface of a similar material. The Arrhenius plot of their conductivity values yielded an activation energy of 41.6 ± 1.7 kJ/mol [31], which was 1.5 times higher than that of the ^7^Li exchange NMR in this work. This difference could be attributed to the different nature of the two activation energies (Figure 7b). The ^7^Li exchange NMR probed the local exchange process between two Li^+^ ions. Therefore, the activation energy of the ^7^Li exchange NMR intrinsically reflected the energy requirement of the Li^+^ local motion. The ionic conductivity measurement required long-range transport of Li^+^ ions. The activation energy from the temperature-dependent conductivities thus reflected the energy requirement of the long-range transport of the Li^+^ ions, which was influenced not only by the local mobility of Li^+^, but also by many other factors such as the pathway of the Li^+^ long-range transportation and the resistance between the electrode and the electrolytes. The different activation energies thus indicated the presence of the additional energy barriers from a local Li^+^ motion to the long-range Li^+^ transportation.

These observations hinted at a possible upper limit for the ionic conductivity of the PEO/Li^+^ complex. The conduction of the PEO/Li^+^ complex was correlated with the Li^+^ motions, which were strongly coupled to the segmental motions of the PEO chain because of the coordination between the Li^+^ ions and the PEO segments. Therefore, the mobility of the PEO segment was critical to the Li^+^ motions and thus the conductivity. Here, the PEO/Li^+^ complexes formed on the surfaces of PEO crystals and consisted of a high content of chain ends, which could greatly facilitate the mobility of PEO segments. The enhanced segmental mobility in turn strongly improved the mobility of the coordinated Li^+^ ions and thus the material conductivity. The fast exchange dynamics of the Li^+^ ions (38 s^−1^ at 305 K) and the relatively low activation energy of the Li^+^ dynamics (28.7 ± 2.3 kJ/mol) concurred with this conclusion. However, the ionic conductivity at room temperature was still 10^−6^ S/cm, even in this special PEO/Li^+^ complex where the segmental mobility was very high because of the highly concentrated chain ends. This conductivity value indicated that the conductivity enhancement via simply increasing the segmental mobility of PEO chain in the PEO/Li^+^ complex had an upper limit. To make a breakthrough on the conductivity, one should enhance the segmental mobility of the PEO/Li^+^ complex. Other approaches are also needed including those used in the ceramic lithium superionic conductors such as attenuating the coordination strength between the Li^+^ ions and the ligands and/or creating a long-range ordered pathway for the Li^+^ ion transportation.

## 4. Conclusions

In summary, we prepared several PEO/Li^+^ complex samples by immersing PEO flakes in Li salt solvents. By using this preparation method, the Li^+^ were concentrated on the surfaces of PEO crystals to form the PEO/Li^+^ complexes. We then used ^13^C and ^7^Li NMR to study the structure and dynamics of the special PEO/Li^+^ complexes. We identified the signal of the Li^+^ ions around the polymer chain ends and the signal of the Li^+^ ions in proximity to the amorphous polymer chain segments. We used 2D ^7^Li-^7^Li exchange NMR to discover experimentally the presence of the exchange dynamics between the two types of Li^+^ ions, which were ascribed to the basic molecular process of the Li^+^ transportation on the surfaces of PEO crystals. We measured the activation energy of the Li^+^ exchange process and found that the activation energy was 1.5-fold lower than that from the conductivity measurement, indicating the presence of additional energy barriers from the local Li^+^ motions to the long-range transportation of the Li^+^ ions. Based on these observations, we discussed the influence of the chain mobility on the material conductivity. In the PEO/Li^+^ complexes, enhancing the conductivity via simply increasing the segmental mobility of PEO chain had an upper limit. To make further breakthroughs on conductivity, one could borrow ideas from ceramic lithium superionic conductors where the conductivity has been significantly enhanced by attenuating the coordination strength between the Li^+^ ions and the ligands, as well as creating a long-range ordered pathway for Li^+^ ion transport. 

## Figures and Tables

**Figure 1 polymers-12-00391-f001:**
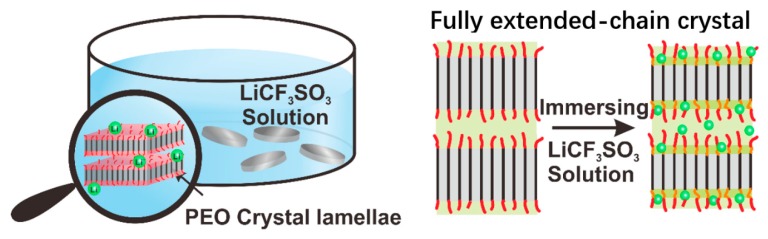
Schematic of the immersion process and Li^+^ distribution. The orange lines and red lines denote the PEO amorphous chain segments and chain ends, respectively. The black lines denote the crystalline PEO chains, and the green balls denote the Li^+^ ions.

**Figure 2 polymers-12-00391-f002:**
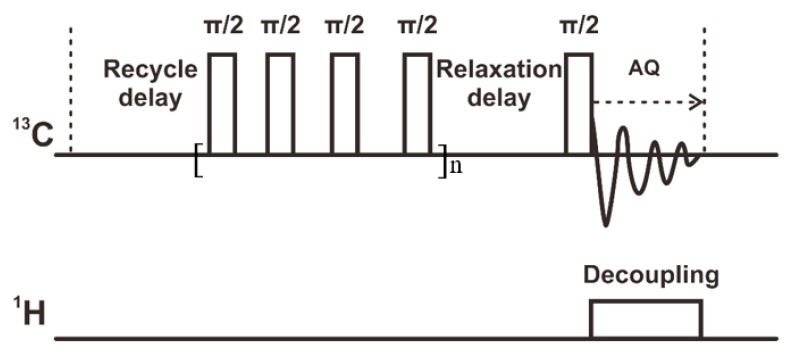
The modified single excitation pulse sequence. The recycle delay was set to 5 s. The relaxation delay was set to a suitable value to select the amorphous signals with a very short relaxation time. AQ: signal acquisition.

**Figure 3 polymers-12-00391-f003:**
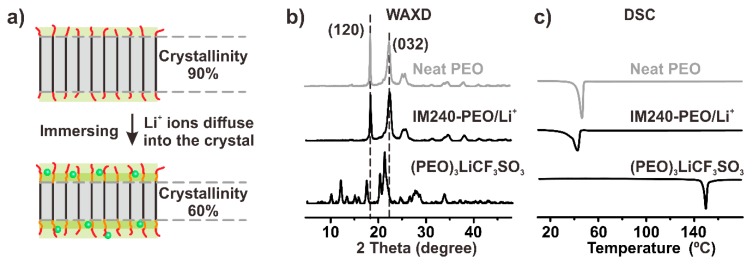
(**a**) Schematic illustrating the immersion process and the Li^+^ ions diffusing into the crystal. The orange lines and red lines denote the PEO amorphous chain segments and chain ends, respectively. The black lines denote the crystalline PEO chains, and the green balls denote the Li+ ions. (**b**) The WAXD patterns of neat PEO, IM240-PEO/Li^+^, and (PEO)_3_LiCF_3_SO_3_. The experimental temperature is room temperature. (**c**) The DSC curves of neat PEO, IM240-PEO/Li^+^, and (PEO)_3_LiCF_3_SO_3_. The samples were heated from 10 to 180 °C at 10 °C/min.

**Figure 4 polymers-12-00391-f004:**
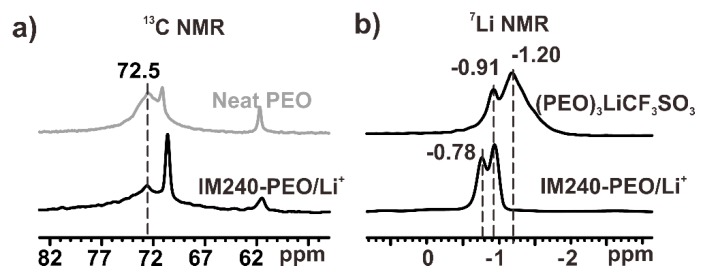
(**a**) The ^13^C NMR spectra of neat PEO and IM240-PEO/Li^+^. (**b**) The ^7^Li NMR spectra of (PEO)_3_LiCF_3_SO_3_ and IM240-PEO/Li^+^. The temperature in the NMR experiments was 300 K.

**Figure 5 polymers-12-00391-f005:**
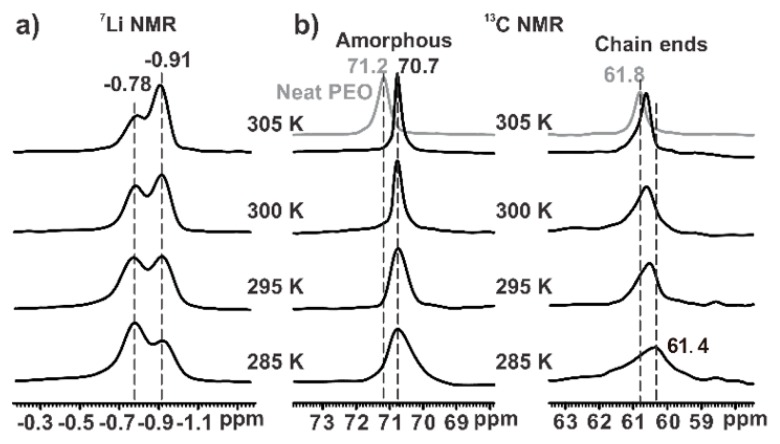
The temperature-dependent (**a**) ^7^Li and (**b**) ^13^C single pulse excitation spectra of IM240-PEO/Li^+^. The ^7^Li NMR spectra were acquired using a recycle delay of 150 s to ensure the full relaxation of the signals. The ^13^C NMR spectra were acquired using a modified single excitation pulse sequence (see Figure 2). The relaxation delay was 0.05 s to suppress the crystalline signal. The experimental temperature ranges from 285 to 305 K. For comparison, the ^13^C NMR spectra of neat PEO are shown.

**Figure 6 polymers-12-00391-f006:**
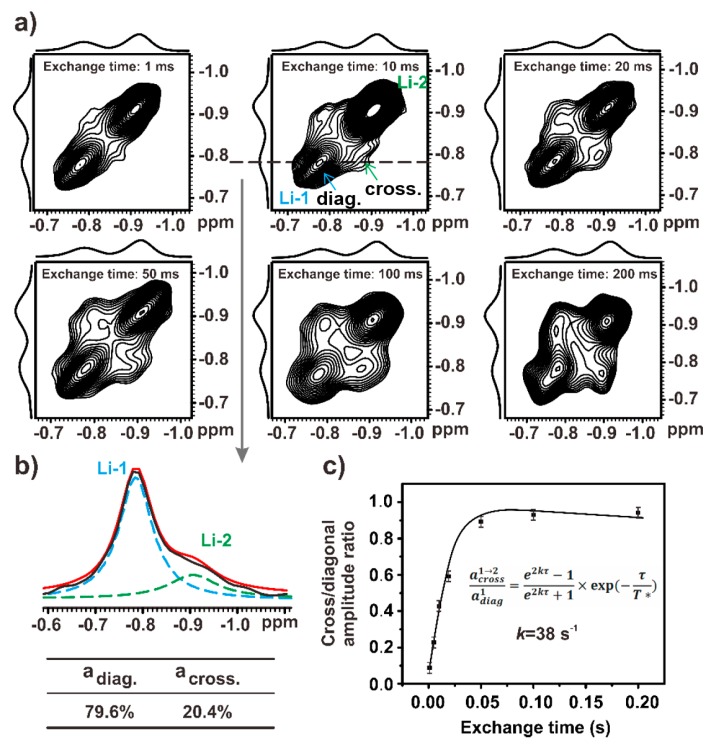
(**a**) The 2D ^7^Li–^7^Li exchange spectrum of IM240-PEO/Li^+^. This spectrum was acquired using an exchange time of 1 to 200 ms at 305 K. (**b**) The cross-section spectrum extracted at δ = −0.78 ppm from 2D exchange ^7^Li–^7^Li spectra of IM240-PEO/Li^+^ with an exchange time of 10 ms (black line). The blue and green dotted lines denoted the fitting peaks of Li-1 and Li-2, respectively. The red line denoted the fitting cross-section spectrum. (**c**) The fitting curve between the ratio of the cross/diagonal peak and exchange time.

**Figure 7 polymers-12-00391-f007:**
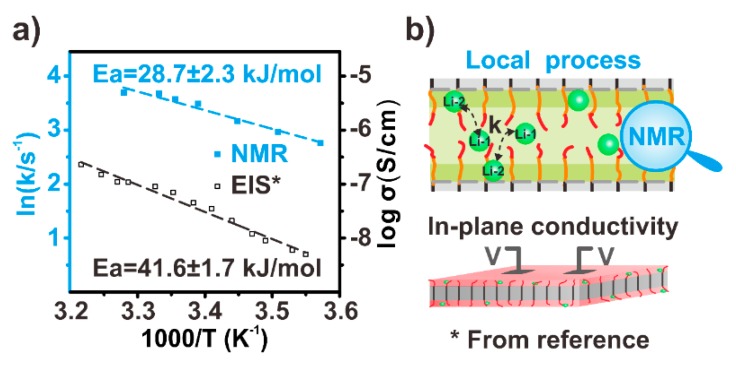
(**a**) Blue square: the Arrhenius plot of the exchange rates obtained from the temperature-dependent 2D ^7^Li–^7^Li exchange NMR. Black open square: the Arrhenius plot of the conductivities reprinted (adapted) with permission from [31]. Copyright (2014) American Chemical Society. (**b**) Top: schematic of the local exchange process of Li^+^ ion. Bottom: the conductivity measurement of the crystal surface with a similar material as Li et al. (reprinted (adapted) with permission from [31]. Copyright (2014) American Chemical Society.). The orange lines and red lines denote the PEO amorphous chain segments and chain ends, respectively. The black lines denote the crystalline PEO chains, and the green balls denote the Li^+^ ions. The pink flat denotes the surface of PEO crystal.

**Table 1 polymers-12-00391-t001:** The ratio of EO: Li^+^ in the samples with three different immersion (IM) times. PEO, polyethylene oxide.

Immersing time	IM105-PEO/Li^+^	IM240-PEO/Li^+^	IM390-PEO/Li^+^
EO:Li^+^ in IM-PEO/Li^+^	10.8:1	5.9:1	5.4:1
EO:Li^+^ in amorphous	3.2:1	2.4:1	2.3:1

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
