# Peer review of "Probing the Dynamics of Li+ Ions on the Crystal Surface: A Solid-State NMR Study"

_polymers, 2020, doi:10.3390/polym12020391_

Round 1

Reviewer 1 Report

Comments on Manuscript (ID polymers-689812)

“Probing The Li+ 2 Ions around Polymer Chain Ends in Polyethylene Oxide/Li 3 + Complex - A Solid State NMR 4 Study”

by Bi-Heng Wang, Tian Xia, Qun Chen and Ye-Feng Yao submited to “Polymers. Polymer Analysis”

The submitted manuscript reports on experimental studies related to solid electrolytes for lithium batteries. The authors applied several advanced techniques to characterize previously prepared polyethylene oxide (PEO) samples with various content of lithium ions. An interesting result on mobility and exchange of two different pools of lithium ions was obtained from 7Li NMR studies in the solid state. Additional insight was made from carbon-13 studies of the same samples. This study could broaden our knowledge about cation dynamic in polymer matrices.

I suggest a minor revision of this submission, mainly related to brushing up the language (the entire text is well written) and correction of several misprints.

In several cases, comma is missing before the word “too” – it should be   “xxx, too.” This is visible in page 2, line 47: “chain ends too.”

Page 3, line 81, better change “to insure” to “to ensure”.

Besides, why in Table 1 is used EO instead of PEO? In addition, please explain how you determine the ratio Lithium cation/PEO.

It would be also nice to include some information on barrier height for the process of lithium ion hopping between different coordination sites (see for example old theoretical/DFT works of Larry Curtiss).

Author Response

Please see the attched file.

Reviewer 2 Report

 The reviewed manuscript deals with an interesting topic of Li+ localization and mobility in polyethylene oxide materials. I find that the main experimental NMR observations and the techniques used to obtain the spectra are sound. The results of 7Li-7Li exchange experiments are especially nice. As with every publication, the interpretation of the primary experimental observations could be debated (see some of my major points below), but in my view this work deserves to be published in Polymers after a minor revision, in which the following should be taken into account:

Major points:
- Which arguments could be provided to justify the cartoons in Figures 1 and 2 showing the loss of PEO crystallinity upon immersion in LiCF3SO3 solution as a monotonous gradient of Li+ concentration from the surface into the bulk of the material? Alternatively, Li+ ions might penetrate the crystal through defects and by forming sort of channels. Please elaborate to increase the clarity.
- In any scenario, it is desirable to discuss the state of CF3SO3- anions. Do they penetrate the bulk of the material as well? Do they cover the surface of the lamellar fragments thus preventing the deep Li+ penetration? Do the results depend on the type of counter anion?
- The preparation and the characterization of the sample labelled (PEO)3LiCF3SO3 should be given. What was the molecular weight of the PEO in this sample? How the stoichiometry was determined? Which evidence can be given for the less mobile Li+ ions being indeed inside the crystal phase (see Figure 4b and its discussion in the text)?
- I believe that the assigment of the 7Li NMR signals in the spectra of PEO/Li+ samples would be incomplete without consideration of the signals that uncoordinated LiCF3SO3 give (so to say, "free Li+"). Were the spectra of free Li salt measured? Could it be the origin of the signal at -0.78 ppm?
- Does the estimated 7Li-7Li exchange rate fit the line widths of the signals at -0.78 and -0.91 ppm (Figure 4b)?
- Section 1. The final part of Section 1 is essentially repeating the Abstract. I think that listing the main results in Introduction should be avoided.

Minor points:
- Abbreviation SPE is confusingly used for Solid Polymer Electrolyte (see f.e. Page 1) and Single Pulse Excitation (see f.e. Page 4).
- The precision of the integrals in Figure S4 seems to be exagerrated, considering the given S/N ratio.
- I suspect that all things considered (S/N ratios, deconvolution algorithm, selected 10 ms exchange time as some of the systematic error sources), the precision of the reported rate constant of 38.1 s-1 is too high. I advice to reduce it to 38 s-1 or even to ca. 40 s-1.
- A small number of typos should be corrected, such as carton, indictes, T1 (should be subscripted), 150.11MHz (space is missing).

Reviewer 3 Report

The article (Manuscript Number polymers-689812) describes the solid-state NMR study of the Li+ ions around the PEO chain ends in the PEO/Li+ complex. The authors prepared the PEO/Li+ complex samples by immersing PEO flakes in Li salt solvents. As they stated the formed PEO/Li+ complexes locating on the surface of the PEO crystal lamellae contain a high amount of chain ends. The Li+ exchange process was considered as the basic molecular process of Li+ transportation in the surface of PEO crystal lamellae. Undoubtedly this research study has brought any interesting results. The authors did not present a specific theme in the article and the potential “real” application of such a procedure. On the whole, the manuscript is not well-written, English could be improved in certain parts. The overall originality of the concept used here is not very high, therefore this method is not making the study useful for the respective field.

Additional comments:

The abstract needs to be well written with future prospects of the work and describe in short the concept of applied techniques. There is a lack of important references related to the used approach in the Introduction. Furthermore, the introduction should be worked out - so as to show the full state of knowledge on this topic. More detailed results discussion should be provided. The chapter appears to be a collection of data from research papers, however, the author’s self-opinion is of importance while drafting a chapter of this type. The conclusion reflects an overall summary of the field with further extension and includes future prospective - I would suggest clarifying this section.

I do not recommend publication of this paper in Polymers.

Author Response

Please see the attched file.

Round 2

Reviewer 3 Report

The article (Manuscript Number polymers-689812) describes the solid-state NMR study of the Li+ ions around the PEO chain ends in the PEO/Li+ complex. The authors prepared the PEO/Li+ complex samples by immersing PEO flakes in Li salt solvents.

Undoubtedly this research study has brought any interesting results. The authors did not present a specific theme in the article and the “real” potential application of such a procedure.

On the whole, the manuscript is not well-written, English could be improved in certain parts. The overall originality of the concept used here is not very high, therefore this method is not making the study useful for the respective field.

I do not recommend publication of this paper in Polymers.